# Evaluation of High-Speed Handpiece Cutting Efficiency According to Bur Eccentricity: An In Vitro Study

**Duk-Yeon Kim [1], Keunbada Son [1,2] and Kyu-bok Lee [1,3,*]**

1   Advanced Dental Device Development Institute, Kyungpook National University, Daegu 41940, Korea
2   Department of Dental Science, Graduate School, Kyungpook National University, Daegu 41940, Korea
3   Department of Prosthodontics, School of Dentistry, Kyungpook National University, Daegu 41940, Korea
*   Correspondence: kblee@knu.ac.kr; Tel.: +82-053-600-7674

**Abstract:** This study aimed to evaluate the correlation between the cutting efficiency and bur eccentricity of high-speed handpieces. The prepared lithium disilicate samples were digitized using a 3D model scanner (reference model, RM) (n = 45), and the lithium disilicate samples were cut using three high-speed handpieces. To evaluate the cutting efficiency, the cut lithium disilicate sample was digitized (cutting model, CM), and the RM and CM were superimposed using a 3D analysis software. Bur eccentricity of the high-speed handpieces was measured using dedicated equipment. Statistical analyses were performed using an analysis software. The statistical differences in pairwise comparisons ($\alpha = 0.05$) were analyzed using the Kruskal–Wallis and post hoc tests. The S-max M600 obtained a cutting efficiency of 6.13 $mm^3$. TG-98 and TRAUS ATN-400 showed similar efficiencies of 2.914 and 3.05 $mm^3$, respectively. There was a significant difference in the cutting efficiency of the S-max M600 compared with TG-98 and TRAUS ATN-400 ($p < 0.001$). S-max M600 had an eccentricity of 3.507 $\mu m$. TG-98 and TRAUS ATN-400 had eccentricities of 5.99 and 7.767 $\mu m$, respectively. There were statistically significant differences in the eccentricity among all the high-speed handpieces ($p < 0.001$).

**Keywords:** cutting efficiency; bur eccentricity; high-speed handpiece; dentistry

## 1. Introduction

Dental high-speed handpieces are the most used tools for cutting and drilling teeth and for most prosthetic therapy. The air turbine handpiece, called the airotor, was developed by John Borden in 1957 and reached bur speeds of 250,000 rpm [1,2].

Though the configuration and driving system of the high-speed handpiece were developed in the 1950s, it is still the most commonly used tool to restore dental prostheses. Therefore, there is still a lot of ongoing research on this appliance. Recent studies mainly focused on evaluating the cutting performance of handpieces with different restorative materials [1,2] and their driving forces, such as air or electricity [3,4], and they performed assessments to reduce noise and vibration [5]. However, few studies have explored the cause of performance differences among the high-speed dental handpieces. In some studies, the performance differences were attributed to bur speed, vibration, and driving forces [6–8].

Bur eccentricity could be one of the causes of the performance difference of high-speed handpieces. Eccentricity is the extent of deviation of the bur from the axis when it rotates about the central axis [9]. Eccentricity is an important factor in the area of industrial drills, although it is less important when evaluating the performance of dental high-speed handpieces. Currently, various studies have been conducted on causes and solutions of eccentricity for industrial drills [10–13]. Eccentricity is a process

based on a new concept, where the turning of parts is performed by a milling tool, regardless of its rotational symmetry. It offers many advantages, such as better chip handling, better surface quality, a higher rate of material removal, and improved tool life due to the interrupted nature of the process as compared to the conventional turning method. Orthogonal, axial, or tangential configurations defined by the arrangement of the tool–workpiece axis can be used during the eccentricity process. Of these, orthogonal eccentricity, in which the tool axis is perpendicular to the workpiece axis, is the most preferred configuration that has been studied. While turning, the operator can define several non-conventional cutting parameters to increase productivity [14]. Bur eccentricity should not be overlooked in the field of prosthetic dentistry where precision work is required. The higher the bur eccentricity, the lower is its cutting accuracy, which can increase the time and expense of dental treatment for the patient and can lead to unnecessary tooth cutting. The international standard for the eccentricity of high-speed dental handpieces has been set at 0.03 mm [15]. However, the effect of eccentricity on the cutting efficiency of high-speed handpieces has not been studied.

The purpose of the present study was to analyze the correlation between the cutting performance and bur eccentricity of three high-speed dental handpieces. To this end, a null hypothesis was set as follows: There is no correlation between the cutting efficiency and bur eccentricity of the three types of high-speed handpieces.

## 2. Materials and Methods

Figure 1 shows the procedure of this study. A pre-crystallization lithium disilicate glass block (IPS e.max CAD, Ivoclar Vivadent, Schaan, Liechtenstein) with a hardness of 5800 MPa and an elastic modulus of 95 GPa, similar to a human tooth with a hardness of 5700 MPa and an elastic modulus of 86.4 GPa, was used [16]. A total of 45 specimens were prepared using a milling machine (Aegis HM, DDS, South Korea). The 15 specimens were used in each of the three handpieces. Pilot experiments were conducted five times to determine the sample size and 10 samples were calculated using a power analysis software (G*Power v3.1.9.4, Heinrich Heine University, Düsseldorf, Germany) (effect size = 92.09%; actual power = 99.30%; power = 99%; $\alpha$ = 0.05). The results indicated that this study needed at least N = 10 subjects to ensure a power >99%. The actual power achieved with this N (99.30%) is slightly higher than the requested power. To increase power, the number of samples presented in this study was determined to be 15. The specimens were prepared to be 12 mm in length, 8 mm in width, and 6 mm thick. The prepared lithium disilicate sample was then digitized as a virtual reference model using a 3D model scanner (E1, 3Shpae, Copenhagen, Danmark). The virtual model was extracted as a stereolithography (STL) file (Figure 2).

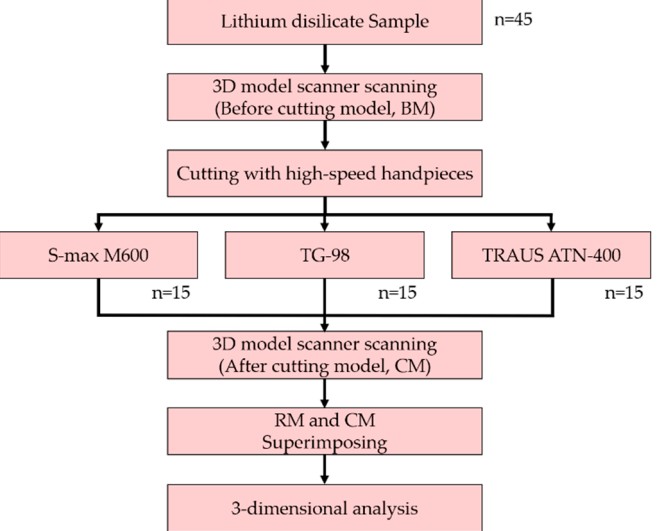

**Figure 1.** Experimental design for evaluating the cutting efficiency.

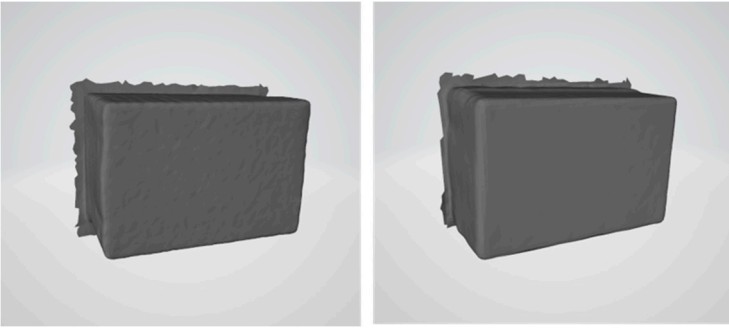

**Figure 2.** 3D virtual model of lithium disilicate reference model.

An experimental custom-made device that was previously referred to in some studies [7,17] was used to hold the high-speed handpiece and specimen in the cutting test. The device consists largely of three parts. The first part is for loading wherein the user is configured to implement a load using the handpiece in the patient's mouth. The second is to fix the specimen to be erected and to fix the sample to the same position during repeated testing. The third part is for fastening the high-speed handpieces and is designed to be fixed in the same position when the tests are repeated after replacing the handpiece. For holding repeatability of the specimen and high-speed handpieces, holding was carried out using a right-angle gauge so that the bur of the high-speed handpiece and specimen were vertical, and the end of the bur was positioned at the end of the specimen. The same procedure was carried out for all the specimens (Figure 3). The three high-speed handpieces were S-max M600 (NSK, Tokyo, Japan), TG-98 (W&H, Bürmoos, Austria), and TRAUS ATN-400 (SAESHIN, Daegu, South Korea). The handpieces were connected to a unit chair (Taurus Sante, Alsafa, Pakistan). The specimen and high-speed handpiece were fixed in a holder, and subsequently, the diamond bur (881G.FG.016, Jota, Rüthi, Switzerland) was positioned with the fixed specimen at the same position. Fifteen specimens were cut with each of the three high-speed handpieces. The handpieces were operated at 300,000 rpm with a cutting force of 0.9 N for 15 s. For this cutting procedure, one high-speed handpiece per manufacturer was used, and a new bur was used for each specimen. The procedure for lubrification of the high-speed handpiece was not performed because the test time was short enough to proceed without lubrification. Furthermore, the heat which generated on the specimen while cutting was cooled by the cooling water injected from the high-speed handpiece. After cutting for each specimen, a five minute break between each experiment was taken to cool down the heat of the high-speed handpiece.

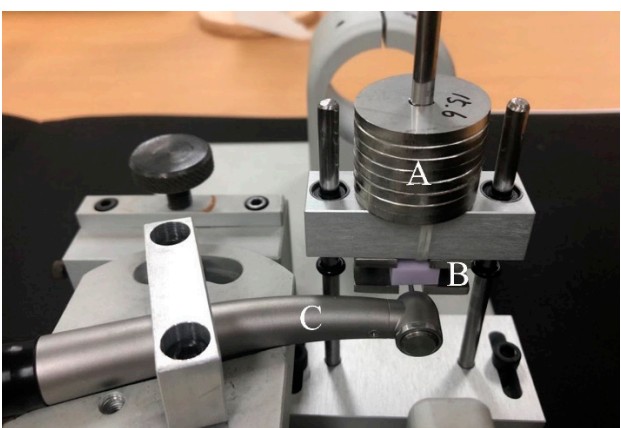

**Figure 3.** The experimental custom-made device: **A.** weight for constant load, **B.** specimen, **C.** high-speed handpiece.

A virtual cutting model was obtained in the same way as the virtual reference model using a 3D model scanner (Figure 4).

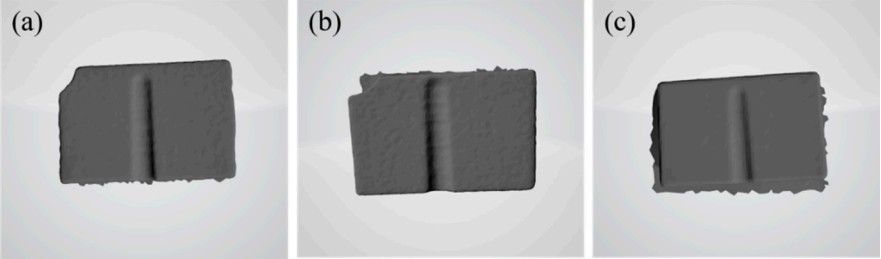

**Figure 4.** 3D virtual model of the lithium disilicate model after cutting using (**a**) S-max M400, (**b**) TG-98, (**c**) TRAUS ATN-400.

The 3D analysis program used in this study was Geomagic Company's 3D inspection software (Geomagic control X, 3D Systems Inc, Rock Hill, SC, USA). The reference model (RM) and cutting model (CM) were superimposed using a 3D inspection software, and then the volume of the deleted part was calculated. When superimposing the STL file that scanned before and after cutting with the 3D analysis software, the area of the cut part can be confirmed, and the volume for this area was calculated by one investigator (K.S.).

A bur eccentricity tester (OPTECH-RI-V, Union Tool, Tokyo, Japan) was used to measure the eccentricity of the high-speed handpieces. The experiment parameters were set at 0.3 MPa of pressure, 400,000 rpm, and 6 mm of bur insertion length.

Statistical analyses were performed using an analysis software (SPSS Statistics 23, IBM, Armonk, NY, USA). The pairwise comparison method was used because there was no normal distribution among groups ($\alpha = 0.05$) and analyzed using the Kruskal–Wallis and post-hoc tests.

## 3. Results

The S-max M600 obtained a cutting efficiency of 6.13 mm$^3$, which was the highest among the three high-speed handpieces, while TG-98 and TRAUS ATN-400 showed similar efficiencies of 2.914 and 3.05 mm$^3$, respectively. Thus, there was a significant difference in the cutting efficiency of the S-max M600 compared with TG-98 and TRAUS ATN-400 ($p < 0.001$; Figure 5).

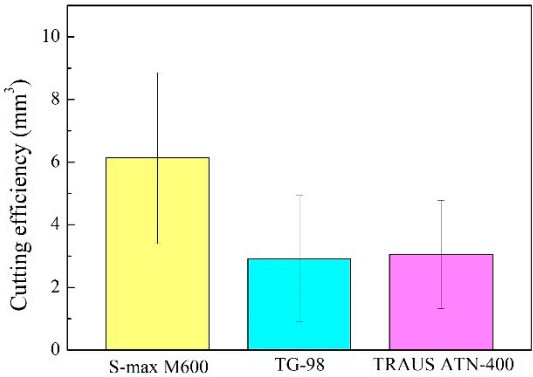

**Figure 5.** Cutting efficiencies of the three kinds of high-speed handpieces.

S-max M600 had an eccentricity of 3.507 μm, which was the smallest among the three high-speed handpieces, while TG-98 and TRAUS ATN-400 had eccentricities of 5.99 and 7.767 μm, respectively. Thus, there were statistically significant differences in eccentricity among the three high-speed handpieces. The p-value for each high-speed handpiece is $p < 0.001$ (Figure 6).

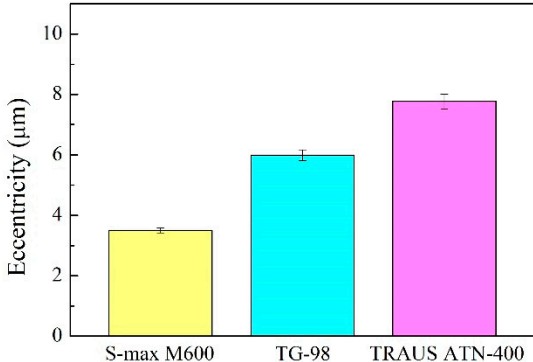

**Figure 6.** Eccentricities of the three kinds of high-speed handpieces.

A comparison of the cutting efficiency and eccentricity showed an inverse correlation, that is, the lower the eccentricity, the higher the cutting efficiency (Figure 7). Statistically, since *p* = 0.011, there is a significant correlation. Further, it has a Pearson correlation coefficient of R = −0.375, which clearly indicates an inverse relationship. S-max M600 had the lowest eccentricity and the highest cutting efficiency, while TG-98 and TRAUS ATN-400 had a lower cutting efficiency due to the increased eccentricity.

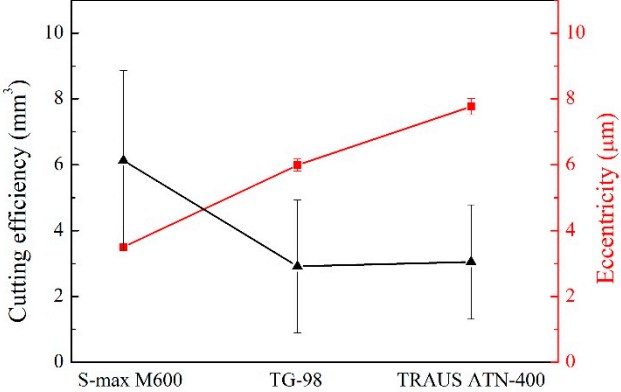

**Figure 7.** Correlation between cutting efficiency and bur eccentricity.

## 4. Discussion

Although studies have mentioned the bur eccentricity of dental high-speed handpieces [8,18], no studies have evaluated the cutting efficiency of high-speed handpieces compared to the bur eccentricity. In this study, the correlation between bur eccentricity and cutting performance of three high-speed handpieces was measured.

The cutting performance of three dental high-speed handpieces was evaluated and S-MAX M600 showed a higher performance that was statistically significant compared to TG-98 and TRAUS ATN-400. On the other hand, S-MAX M600 showed the lowest bur eccentricity, while TG-98 and TRAUS ATN-400 showed a progressively higher bur eccentricity. The findings indicate that bur eccentricity and cutting efficiency tend to be inversely proportional to each other, that is, the lower the bur eccentricity, the higher the cutting efficiency. Since the rotational speed, bur type, type of cut specimen, and size of the load were fixed in the study, the eccentricity of the high-speed handpieces probably affects the cutting efficiency. Eccentricity influences the cutting efficiency because it changes the tooling case and the local cutting conditions (cutting speed and angle), which implies that there is no cutting position at one or several cutting edges. It also degrades the surface quality and reduces the stability of the process. In addition, the centrifugal force generated by eccentricity may result in the failure of high-speed handpieces [19–21].

The vibration caused by bur eccentricity may cause abrasion and fatigue of the bur, as well as of the connected bearings, shafts, and gears, and reduces the machine life, eventually rendering it defective [11]. Therefore, eccentricity should be carefully considered when selecting a high-speed handpiece. Although our findings suggest that S-MAX M400 showed the best performance, all three high-speed handpieces were clinically acceptable.

Therefore, when selecting high-speed handpieces, the design, price, brand, maximum bur speed, vibration, and noise must be considered. However, more specific criteria are needed for the selection of high-speed handpieces used in the field of prosthetic dentistry.

The present study has limitations. Using the ceramic block with similar mechanical strength to human teeth without studying the actual oral environment, the experimental conditions did not match the actual oral environment. In future studies, in vivo or ex vivo studies should be conducted. In addition, one type of diamond bur was used, which cannot identify the effects of various types of burs. Further research also needs to be done to evaluate the effects of various types of burs.

## 5. Conclusions

Based on the findings of this in vitro study, the following conclusions were drawn:

1.  Since the bur eccentricity of the high-speed handpiece was found to affect the cutting efficiency, the null hypothesis was rejected.
2.  The cutting efficiency and bur eccentricity of the high-speed handpiece are inversely related.

**Author Contributions:** Conceptualization, D.-Y.K.; Methodology, K.-b.L.; Validation, K.-b.L.; Formal Analysis, D.-Y.K.; Investigation, D.-Y.K.; Data Curation, K.S.; Writing—Original Draft, D.-Y.K.; Visualization, S.K.; Supervision, K.-b.L.; Project Administration, K.-b.L.

**Funding:** This research was supported by the Technology Innovation Program (or Industrial Strategic Technology Development Program, 10077743). The development of the handpiece design for the air turbine and root canal treatment was funded by the Ministry of Trade, Industry & Energy (MOTIE, Korea) and Korea Institute for Advancement of Technology (KIAT) through the National Innovation Cluster R&D program (P0006691).

**Acknowledgments:** The authors thank the researchers of the Advanced Dental Device Development Institute, Kyungpook National University, for their time and contributions to the study.

**Conflicts of Interest:** The authors declare no conflicts of interest. The funders had no role in the design of the study; in the collection, analyses, or interpretation of the data; in the writing of the manuscript, or in the decision to publish the results.

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
