# Peer review of "Evaluation of High-Speed Handpiece Cutting Efficiency According to Bur Eccentricity: An In Vitro Study"

_applsci, doi:10.3390/app9163395_

Round 1

Reviewer 1 Report

Dear author, I read the manuscript. The research could be of interest for readers, nevertheless, there are several issues to be fixed, most in the materials and methods section. Please refer to the attached PDF with step-by-step suggestions. In general, the methods must to be clearly described. I suggest to follo the CRIS guidelines to improve the alit of the manuscript. Last but not the least, please discuss possible clinical relevance of this research, for further study.

https://www.researchgate.net/project/CRIS-guidelines

Author Response

We are grateful to the reviewers for their critical comments and useful suggestions that have helped us to greatly improve our paper. As indicated in the following responses, we have reflected all these comments in the revised version of our paper. Furthermore, we have had the manuscript checked by a professional English editing service.

Reviewer #1

P.1, line 15: “to elucidate”: suggest “evaluate”

Response: Thank you for your comment. We have changed “evaluate”.

P.1, line 21: “were overlapped”: suggest “superimposed”

Response: Thank you for your comment. We have changed “superimposed”.

P.1, line 28, 30: Report the P values

Response: Thank you for your comment. We have reported the expect P value.

“There was a significant difference in the cutting efficiency of the S-max M600 compared with TG-98 and TRAUS ATN-400 (P =0.667). S-max M600 had an eccentricity of 3.507 μm. TG-98 and TRAUS ATN-400 had ccentricities of 5.99 and 7.767 μm, respectively. There were statistically significant differences in the ccentricity among all the high-speed handpieces. (P<0.001)

P.1, line 37: “restorations”: suggest “therapy”

Response: Thank you for your comment. We have changed “therapy”.

P.2, line 54: Provide the reference about “Eccentricity is the extent of deviation of the bur from the axis when it rotates about the central axis.”

Response: Thank you for your suggestion for improving the quality of the manuscript. We have provided the reference for explanation [9] Takashi, M.; Shoichi, T.; Cutting force model in milling with cutter runout, Procedia, 2017, 58, 566-571

P.2, line 69: Report only the aim of the research and the hypothesis. Move other informations in the materials and methods, and the results of the correlation, in the results section.

Response: Thank you for your suggestion for improving the quality of the manuscript. We have modified the contents. “The purpose of the present study was to analyze the correlation between the cutting performance and bur eccentricity of three high-speed dental handpieces. To this end, a null hypothesis was set, as follows: There is no correlation between the cutting efficiency and bur eccentricity of the three types of high-speed handpieces.”

P.2, line 77: Please refer to CRIS guidelines

Response: Thank you for your suggestion for improving the quality of the manuscript. We have added a substitute for sapme handling, sample size calculation, and statistical analysis, referring to the CRIS guideline.

P.3, line 80: All the paragraph seem to be a repetition of the text.

Response: Thank you for your comment. We have modified that paragraph simply.

P.3, line 83: “overlapping”: suggest “superimposition”

Response: Thank you for your comment. We have deleted the paragraph that include “overlapping”

P.3, line 85: Introduce the study design. Did the authors perform a sample size calculation? Also report how the samples were allocated, as well as, blinding procedure.

Response: Thank you for your comment. We have added the content to explaining the sample size calculation. “15 specimens were used in each of the three handpieces. Pilot experiments were conducted five times to determine the sample size and 10 samples were calculated using a power analysis software (G*Power v3.1.9.4, Heinrich Heine University, Düsseldorf, Germany) (actual power = 99.30%; power = 99%; α = 0.05). To increase power, the number of samples presented in this study was determined to be 15.”

P.3, line 95: What about calibration of the device?

Response: Thank you for your comment. We have added the content to explaining the calibration of the device. “For holding repeatability of the specimen and high-speed handpieces, holding was carried out using a right-angle gauge so that the bur of the high-speed handpiece and specimen were vertical, and the end of bur positioned at the end of the specimen. Also, the same procedure was carried out for all the specimens.”

P.3, line 103: Did the authors used new burs? Did the authors used new handpiece? Provide information abut the diamond bur changing and the lubrification methods used.

Response: Thank you for your comment. We have added the contents to explaining about the bur, handpieces.

“One high-speed handpiece per manufacturer was used, and a new bur was used for each specimen and the procedure for lubrification of the high-speed handpiece was not performed because the test time was short  to proceed with lubrification.”

P.3, line 106: What about cooling

Response: Thank you for your commnet. We have added the contents to explaining the colling

 “The heat which generated on the specimen while cutting was cooled by the cooling water injected from the high-speed handpiece. After cutting for each specimen, the 5 minute break between each experiment was made to cool the heat of the high-speed handpiece. “

P.4, line 119: Provide more information. How did the volume calculated?

Response: Thank you for your comment. We have added the explaining about volume calculate.

 “When superimposing the STL file that scanned before and after cutting with 3D analysis software, the area of the cut part can be confirmed and the volume for this area is calculated by software.”

P.4, line 120: Define the outcomes of the study. In the abstract authors referred to CUTTING EFFICIENCY

ECCENTRICITY Define each outcome in detail.

Response: Thank you for your comment. The results of the cutting efficiency and eccentricity and correlation results of the cutting efficiency and eccentricity are described in the results chapter.”

P.4, line 123: Please explain why authors performed a pairwise comparison. What is the rationale to use a non-parametric test? In the introduction, authors mentioned correlation between outcomes. Please report.

Response: Thank you for your comment. We have added the explaining for pairwise comparison. “The pairwise comparison method was used because there was no normal distribution among groups.”

P.4, line 133: Report exact p values

Response: Thank you for your comment. We have reported the expect P value. “P=0.667

P.5, line 141: Report exact p values

Response: Thank you for your comment. We have reported the expect P value. “The p-value for each high-speed handpiece is p <0.001.”

P.5, line 146: Report exact p values

Response: Thank you for your comment.  We have reported exact P value and ground “Statistically, since p = 0.011, there is a significant correlation. Also, it has a Pearson correlation coefficient R = -0.375, which clearly indicates an inverse relationship.”

P.6, line 168: suggest “may”

Response: Thank you for your comment. We have changed “eccentricity may cause”.

P.5, line 151: Please also report the limitation of the study: sample size, possible bias, in vitro design...

Response: Thank you for your suggestion for improving the quality of the manuscript. We have added the limitation of this study.

Limitations of this study

Using the ceramic block with similar mechanical strength to human’s teeth without studying the actual oral environment, the experimental conditions did not match the actual oral environment. In future studies, in-vivo or ex vivo studies should be conducted. One type of diamond bur was not used to identify the effects of various types of burs.”

Reviewer 2 Report

dear Authors, 

i have the following comments about the manuscript "EVALUATION OF HIG-SPEED HANDPIECE CUTTING EFFICIENCY ACCORDING TO BUR ECCENTRICITY: AN IN-VITRO STUDY":

the topic is interesting, but the article can be improved in its entirety;

the section ABSTRACT exceeds the 200 words and it doesn't follow the style of structured abstract: it lacks the background.

the section INTRODUCTION doesn't highlight why this study is important;

please specify in the section MATERIALS AND METHODS if the diamond bur are changed each time a specimen is changed;

in the section DISCUSSION, on the basis of what references can you claim the suitability of the statement from the line 68 and 169?

Please describe the references as request in the Instructions for Authors.

Author Response

We are grateful to the reviewers for their critical comments and useful suggestions that have helped us to greatly improve our paper. As indicated in the following responses, we have reflected all these comments in the revised version of our paper. Furthermore, we have had the manuscript checked by a professional English editing service.

Reviewer #2

the section ABSTRACT exceeds the 200 words and it doesn't follow the style of structured abstract: it lacks the background.

Response: Thank you for your comment. We have modified the ABSTRACT to follow the style of structured abstract.

the section INTRODUCTION doesn't highlight why this study is important;

Response: Thank you for your comment. We have provided a supplementary explanation of why this study is important in introduction.

“eccentricity is an important factor in the area of industrial drills, although it is less important when evaluating the performance of dental high-speed handpieces. Currently, various studies have been conducted on causes and solutions of eccentricity for industrial drills.”

[10] C, C, Tsao.; H, Hocheng.; Effect of eccentricity of twist drill and candle stick drill on delamination in drilling composite materials, International Journal of Machine Tools and Manufacture, 2005, 45, 125-130.

[11] M, J, Knight.; F, P, Brennan.; Fatigue life improvement of drill collars through control of bore eccentricity, Engineering Failure Analysis, 1999, 6, 301-319.

[12] D, S, Raj.; L, Karunamoorthy.; Study of the effect of Tool Wear on Hole Quality in Drilling CFRP to Select a Suitable Drill for Multi-Criteria Hole Quality, Materials and Manufacturing Processes, 2016, 31, 587-592.

[13] Investigation of drill pipe rotation effect on cutting transport with aerated mud using CFD approach, Advanced Powder Technology, 2017, 28, 1141-1153.

please specify in the section MATERIALS AND METHODS if the diamond bur are changed each time a specimen is changed;

Response: Thank you for your comment. We have added the contents to explaining about the bur, handpieces.

“One high-speed handpiece per manufacturer was used, and a new bur was used for each specimen and the procedure for lubrification of the high-speed handpiece was not performed because the test time was short  to proceed with lubrification.”

in the section DISCUSSION, on the basis of what references can you claim the suitability of the statement from the line 168 and 169? Please describe the references as request in the Instructions for Authors.

Response: Thank you for your suggestion for improving the quality of the manuscript. We have added references to improve the discussion section.

[11] Fatigue life improvement of drill collars through control of bore eccentricity, Engineering Failure Analysis, 1999, 6, 301-319.

Reviewer 3 Report

This is an original and interesting in vitro study. Certainly appreciate for the efforts. I have following comments and concerns regarding the study. 

1. First of all concerns with this work, is the study methodology. It does not seem clear why 45 specimens were used; if a statistical criteria based on sample size was followed, it should be evidently described.

2. The second concerns relates with the evaluation of the specimen's surface quality after cutting. Did you consider any method? I suggest to consider confocal microscope in order to better appreciate in vitro and in vivo surface evaluation (Gentile E, Di Stasio D, Santoro R, Contaldo M, Salerno C, Serpico R, Lucchese A. In vivo microstructural analysis of enamel in permanent and deciduous teeth. Ultrastruct Pathol. 2015 Apr;39(2):131-4. doi: 10.3109/01913123.2014.960544.)

3. At last, in the aim to improve a wide readership, I recommend authors should mention the potential implications of the in vitro  cutting performance with in vivo effects on oral health, also according to prognostic tools (Pannone G, Sanguedolce F, De Maria S, Farina E, Lo Muzio L, Serpico R, Emanuelli M, Rubini C, De Rosa G, Staibano S, Macchia L, Bufo P. Cyclooxygenase isozymes in oral squamous cell carcinoma:a real-time RT-PCR study with clinic pathological correlations. Int J Immunopathol Pharmacol. 2007 Apr-Jun;20(2):317-24. PubMed PMID: 17624243.)

Overall, I really appreciate the efforts of authors.

Author Response

We are grateful to the reviewers for their critical comments and useful suggestions that have helped us to greatly improve our paper. As indicated in the following responses, we have reflected all these comments in the revised version of our paper. Furthermore, we have had the manuscript checked by a professional English editing service.

Reviewer #3

First of all concerns with this work, is the study methodology. It does not seem clear why 45 specimens were used; if a statistical criteria based on sample size was followed, it should be evidently described.

Response: Thank you for your comment. We have added the content to explaining the sample size calculation. “15 specimens were used in each of the three handpieces. Pilot experiments were conducted five times to determine the sample size and 10 samples were calculated using a power analysis software (G*Power v3.1.9.4, Heinrich Heine University, Düsseldorf, Germany) (actual power = 99.30%; power = 99%; α = 0.05). To increase power, the number of samples presented in this study was determined to be 15.”

The second concerns relates with the evaluation of the specimen's surface quality after cutting. Did you consider any method? I suggest to consider confocal microscope in order to better appreciate in vitro and in vivo surface evaluation (Gentile E, Di Stasio D, Santoro R, Contaldo M, Salerno C, Serpico R, Lucchese A. In vivo microstructural analysis of enamel in permanent and deciduous teeth. Ultrastruct Pathol. 2015 Apr;39(2):131-4. doi: 10.3109/01913123.2014.960544.)

Response: Thank you for your suggestion for improving the quality of the manuscript. This study was conducted to determine the cutting efficiency of the high-speed handpiece by specifying only the volume change of the specimen. The confocal microscope for surface quality evaluation will be reflected in future studies.

At last, in the aim to improve a wide readership, I recommend authors should mention the potential implications of the in vitro cutting performance with in vivo effects on oral health, also according to prognostic tools (Pannone G, Sanguedolce F, De Maria S, Farina E, Lo Muzio L, Serpico R, Emanuelli M, Rubini C, De Rosa G, Staibano S, Macchia L, Bufo P. Cyclooxygenase isozymes in oral squamous cell carcinoma:a real-time RT-PCR study with clinic pathological correlations. Int J Immunopathol Pharmacol. 2007 Apr-Jun;20(2):317-24. PubMed PMID: 17624243.)

Response: Thank you for your suggestion for improving the quality of the manuscript. We have added the limitation of this study.

“Limitations of this study :

Using the ceramic block with similar mechanical strength to human’s teeth without studying the actual oral environment, the experimental conditions did not match the actual oral environment. In future studies, in-vivo or ex vivo studies should be conducted. One type of diamond bur was not used to identify the effects of various types of burs.”

Round 2

Reviewer 1 Report

Dear authors, thanks to provide a new version of the manuscript. A noted that most of the reviewers' suggestions have been considered and the manuscript has been improved. Nevertheless, I still have some concernes most about sample size calculation. Please check.

Author Response

We are grateful to the reviewers for their critical comments and useful suggestions that have helped us to greatly improve our paper. As indicated in the following responses, we have reflected all these comments in the revised version of our paper. Furthermore, we have had the manuscript checked by a professional English editing service.

Reviewer #2

The P Value is not significant. Please check.

Response: Thank you for your accurate comment. As the reviewer notes, we have revised the p value.

Sample size calculation is still not clear. Author stated that pilot study was used to estimate the sample size. Nevertheless, effect size is not reported. Then, what does actual power mean? Please explain and give us information about its calculation.

Response: Thank you for your suggestion for improving the quality of the manuscript. We have carefully considered your comments. We have added the effect size. Actual power means the power of N calculated by the value of power set in software. We have revised the issue that you point out so that the sample size calculation is readily comprehensible to readers.

“The 15 specimens were used in each of the three handpieces. Pilot experiments were conducted five times to determine the sample size and 10 samples were calculated using a power analysis software (G*Power v3.1.9.4, Heinrich Heine University, Düsseldorf, Germany) (effect size = 92.09%; actual power = 99.30%; power = 99%; α = 0.05). The results indicate that this study need at least N = 10 subjects to ensure a power > 99%. The actual power achieved with this N (99.30%) is slightly higher than the requested power. To increase power, the number of samples presented in this study was determined to be 15.”

Why can? The area and volume have been evaluated by an operator? Please report the initials. Was the operator blind?

Response: We very much appreciate the reviewer’s comment and respect the reviewer’s insight. We have revised the issue that you point out.

“When superimposing the STL file that scanned before and after cutting with 3D analysis software, the area of the cut part can be confirmed and the volume for this area was calculated by one investigator (K.S.).

Please move the limitations in the discussion section.

Response: Thank you for your suggestion for improving the quality of the manuscript. As the reviewer notes, we have added text to improve the Discussion section.

The present study has limitations. Using the ceramic block with similar mechanical strength to human’s teeth without studying the actual oral environment, the experimental conditions did not match the actual oral environment. In future studies, in-vivo or ex vivo studies should be conducted. In addition, one type of diamond bur was not used to identify the effects of various types of burs. Further research also needs to be evaluated the effects of various types of burs.